# Phenformin Down-Regulates c-Myc Expression to Suppress the Expression of Pro-Inflammatory Cytokines in Keratinocytes

**DOI:** 10.3390/cells11152429

**Published:** 2022-08-05

**Authors:** Guanyi Liu, Dingyang Li, Liwei Zhang, Qiuping Xu, Dexuan Zhuang, Panpan Liu, Ling Hu, Huiting Deng, Jianfeng Sun, Shuangshuang Wang, Bin Zheng, Jing Guo, Xunwei Wu

**Affiliations:** 1Department of Tissue Engineering and Regeneration, School and Hospital of Stomatology, Cheeloo College of Medicine, Shandong University & Shandong Key Laboratory of Oral Tissue Regeneration & Shandong Engineering Laboratory for Dental Materials and Oral Tissue Regeneration, No. 44-1 Wenhua Road West, Jinan 250012, China; 2Department of Orthodontics, School and Hospital of Stomatology, Cheeloo College of Medicine, Shandong University & Shandong Key Laboratory of Oral Tissue Regeneration & Shandong Engineering Laboratory for Dental Materials and Oral Tissue Regeneration, No. 44-1 Wenhua Road West, Jinan 250012, China; 3Engineering Laboratory for Biomaterials and Tissue Regeneration, Ningbo Stomatology Hospital, Savaid Stomatology School, Hangzhou Medical College, Ningbo 315000, China; 4Department of Pediatrics Dentistry, Department of Preventive Dentistry, School and Hospital of Stomatology, Cheeloo College of Medicine, Shandong University & Shandong Key Laboratory of Oral Tissue Regeneration & Shandong Engineering Laboratory for Dental Materials and Oral Tissue Regeneration, No. 44-1 Wenhua Road West, Jinan 250012, China; 5Cutaneous Biology Research Center, Massachusetts General Hospital, Harvard Medical School, Boston, MA 02129, USA

**Keywords:** skin inflammation, phenformin, cytokines, c-Myc, mTOR

## Abstract

The treatment of many skin inflammation diseases, such as psoriasis and atopic dermatitis, is still a challenge and inflammation plays important roles in multiple stages of skin tumor development, including initiation, promotion and metastasis. Phenformin, a biguanide drug, has been shown to play a more efficient anti-tumor function than another well-known biguanide drug, metformin, which has been reported to control the expression of pro-inflammatory cytokines; however, little is known about the effects of phenformin on skin inflammation. This study used a mouse acute inflammation model, ex vivo skin organ cultures and in vitro human primary keratinocyte cultures to demonstrate that phenformin can suppress acute skin inflammatory responses induced by 12-O-tetradecanoylphorbol-13-acetate (TPA) in vivo and significantly suppresses the pro-inflammatory cytokines IL-1β, IL-6 and IL-8 in human primary keratinocytes in vitro. The suppression of pro-inflammatory cytokine expression by phenformin was not directly through regulation of the MAPK or NF-κB pathways, but by controlling the expression of c-Myc in human keratinocytes. We demonstrated that the overexpression of c-Myc can induce pro-inflammatory cytokine expression and counteract the suppressive effect of phenformin on cytokine expression in keratinocytes. In contrast, the down-regulation of c-Myc produces effects similar to phenformin, both in cytokine expression by keratinocytes in vitro and in skin inflammation in vivo. Finally, we showed that phenformin, as an AMPK activator, down-regulates the expression of c-Myc through regulation of the AMPK/mTOR pathways. In summary, phenformin inhibits the expression of pro-inflammatory cytokines in keratinocytes through the down-regulation of c-Myc expression to play an anti-inflammation function in the skin.

## 1. Introduction

Many skin diseases are related to skin inflammation. Although great progress has been made in the development of new methods for the treatment of skin inflammation in recent years, especially in targeted therapy [1,2,3], there are still great challenges for the treatment of many skin diseases that are obviously related to inflammation, such as psoriasis and atopic dermatitis. Therefore, characterization of the molecular mechanism(s) of skin inflammatory responses and the regulation of skin inflammatory responses will definitely contribute to the prevention and treatment of many inflammatory skin diseases, which has always been a challenge in the field of skin research. Inflammation is an immune response involving immune cells mediated by pro-inflammatory cytokines, which play key roles in maintaining homeostasis in the body [4]. The skin is the first line of defense for the body against the external environment and the outermost epidermal layer of the skin, mainly composed of keratinocytes, is the vanguard of that defense line and plays a key role in immune responses of the skin [5]. Studies have shown that inflammation of the skin is usually caused by various kinds of inflammatory factors secreted by keratinocytes in the epidermis after various external stimuli, which then mobilize immune cells in the skin and interact with them to produce the corresponding immune response [6,7,8,9,10,11,12,13,14]. Furthermore, keratinocytes undergo a highly regulated and programmed differentiation process to form several epidermal layers with distinct levels of cell differentiation, ranging from less differentiated basal cells at the bottom, to moderately differentiated cells in the spinous and granular layers and highly differentiated corneocytes at the top, which play an essential role for the barrier function of the skin to prevent trans-epidermal water loss (TEWL) [15,16,17]. This differentiation program is a sequence of well-regulated cell processes that result from keratinocytes communicating with other skin-resident cells through the production of cytokines that are responsible for the regulation of cellular communications. Cytokines have been shown to play important roles in regulating keratinocyte proliferation and differentiation and deregulated cytokine signaling can result in multiple consequences for the barrier function of the skin, which is observed in many chronic inflammation diseases, including atopic dermatitis (AD) and psoriasis [15,18]. Taken together, we conclude that keratinocytes play a leading role in skin inflammatory responses and are the key cells to study about inflammatory responses in the skin.

Metformin has been the most commonly used drug for the treatment of type II diabetes in recent years. It has recently been found that metformin has a wide range of anti-cancer and anti-aging effects and prolongs life expectancy [19,20,21], which has attracted the attention of researchers. Metformin has been reported to suppress inflammatory responses in different types of cells; for instance, metformin can induce the expression of ATF3 to inhibit the inflammatory response in mouse macrophages [22] and metformin has been shown to suppress the expression of pro-inflammatory factors, such as IL-6 and TNF-α, by downregulating the mTOR signaling pathway in immortalized HaCaT keratinocytes [23]. In human skin, treatment with metformin has been shown to benefit patients by reducing the risk of psoriasis, as well as by improving the disease severity in clinics. Metformin has been reported to inhibit the upregulated expression of psoriasis-related cytokines, such as IL-1β, IL-36γ, CXCL1, CXCL2, CCL20, S100A7, S100A8 and S100A9 in human keratinocytes. However, the role of metformin in regulating the expression of inflammatory cytokines in human keratinocytes or in the skin has still not been fully explored. Phenformin, another biguanide drug, was also used to treat type II diabetes in the early 1970s but the use of phenformin was clinically discontinued at the end of the 1970s due to a high risk of fatal lactic acidosis. However, in recent years, it has been discovered that phenformin has a more efficient anti-tumor function than metformin in multiple types of cancers, such as melanoma [24,25,26]. The main reason for that is that phenformin can be effectively absorbed by tumor cells, while metformin needs to bind a transporter, called an organic cation transporter (OCT), to enter cells [27,28,29,30]. The OCT transporter is highly expressed in the liver, intestine and kidney, but is not expressed at a detectable level in the skin and, therefore, metformin is poorly absorbed by the skin [31]. We recently reported that phenformin plays a tumor-suppressing function by promoting keratinocyte differentiation through regulation of the NFAT (nuclear factor T cells) pathway, which is the master regulator of immune responses [32]. These data suggest that phenformin may play an important role in regulating inflammatory responses in keratinocytes, and, therefore, the aim of this study was to investigate whether phenformin controls skin inflammation and to clarify its underlying mechanism(s).

## 2. Materials and Methods

### 2.1. Isolation and Culture of Primary Human Keratinocytes

The isolation and culture of primary human keratinocytes was performed as previously described [33,34,35]. Briefly, neonatal foreskin tissues were obtained from discarded hospital specimens and were incubated overnight in dispase solution in PBS (2.5 mg/mL) at 4 °C. The next day, the epidermis was peeled off and separated from the dermis, then minced and incubated with 0.05% Trypsin-EDTA (Cat. 25300054, ThermoFisher Scientific, Waltham, MA, USA) at 37 °C for 30 min. The digestion was then stopped with 10% fetal bovine serum (FBS) in Dulbecco’s modified Eagle’s medium (DMEM, Gibco, Grand Island, NY, USA) and the solutions were filtered, centrifuged and washed to obtain the cell pellets. The cell pellets were resuspended in the culture medium K-SFM (Cat. 10744019, Gibco, Grand Island, NY, USA) supplemented with 100 mg/mL streptomycin and 100 U/mL penicillin. Human keratinocytes were cultured at 37 °C in a humidified incubator containing 5% CO_2_. Passage 3 cells were used for all experiments. For compound treatments, cells were seeded in ultralow attachment 6-well plates (Cat. 3471, Corning, Corning, NY, USA) with K-SFM in the presence or absence of different concentrations (0.5, 1.0 or 1.5 mM) of phenformin (Cat. P7045, Sigma-Aldrich, St. Louis, MO, USA), 10 μg/mL Polyinosinic:polycytidylic acid (Poly (I:C)) (InvivoGen, San Diego, CA, USA), 5 μg/mL 12-O-tetradecanoylphorbol-13-acetate (TPA, Cat. P8139, Sigma-Aldrich, St. Louis, MO, USA) or 10 μM MHY1485 (Cat. S7811, Selleck Chemicals, Houston, TX, USA) for the desired times, or AICAR (0.5 or 1.0 mM, Cat. S1802, Selleck Chemicals, Houston, TX, USA) or JQ1 (25 or 50 nM, Cat. S7110, Selleck Chemicals, Houston, TX, USA); detailed conditions are described in the Figure Legends.

### 2.2. Human Skin Ex Vivo Cultures (Skin Organ Cultures)

Skin organ cultures followed a previously described protocol [36]. Briefly, human skin tissues, obtained from discarded hospital specimen samples from plastic surgery, were delivered to the laboratory within 1–1.5 h post-surgery in Williams’ E medium (Cat. W1848, Sigma-Aldrich, St. Louis, MO, USA) on ice. The skin was defatted, cut by a scalpel into small 5 mm pieces, rinsed abundantly with PBS (pH 7.2) and then epidermis up/dermis was placed down into 6-well plates. Skin samples were cultured overnight in a humidified atmosphere (5% CO_2_, 37 °C) in 2 mL serum-free Williams’ E medium. The next day, each piece of tissue was treated topically with 100 μL (100 mg/mL Poly (I:C)) for 2 h, then with a 100 μL aliquot of different concentrations of 1.5, 3.0 or 6.0 mM phenformin for 22 h, after which the tissues were collected for extraction of total mRNA.

### 2.3. Mouse Model for Acute Skin Inflammation

The in vivo acute skin inflammation model induced by TPA was performed as previously described with modification [37]. To study the effect of phenformin on acute skin inflammatory responses, 8-week-old female C57 mice (Charles River, Beijing, China) were randomly divided into two groups (6 mice per group) and were treated with phenformin (150 mg/kg in PBS) or with metformin (250 mg/kg in PBS) or with an equal volume of PBS as a control by oral gavage twice a day for two days. The following day, the right ear of each mouse was topically treated with a single dose of 20 μL TPA (10 μL for each inner and outer surface), which had been dissolved in acetone (100 μg/mL) and the other ear was treated with 20 μL acetone as a negative control. To study the effect of JQ1 on acute skin inflammatory responses, the same strain of mice was randomly divided into two groups (6 mice per group) and treated topically on the inner (10 μL) and outer (10 μL) surfaces of their right ears with 20 μL JQ1 (0.5 mg/ear in DMSO, Cat. S7110, Selleck Chemicals, Houston, TX, USA). The right ears of the other group of mice were treated with 20 μL acetone/DMSO as a control group for 30 min, then were treated with TPA as described above. The ear thickness of each mouse after TPA treatment was measured at different time points using a digital caliper (Mitutoyo Corp., Tokyo, Japan) and photos of ears were taken under the same lighting conditions to show acute inflammatory responses. For further analysis of ear acute inflammatory responses, at 8 h after treatment with TPA, the mice were sacrificed and their ears were collected for histological and immunofluorescence (IF) analyses. These experiments were performed following the guidelines from the Ethics Committee of the Stomatological Hospital Shandong University.

### 2.4. Histology and IF Analysis

Mouse ear tissue samples were frozen after embedding in optimal cutting temperature compound (Tissue-Tek, Torrance, CA, USA), after which 8 µm cryosections were prepared for histological analysis according to specific instructions. The IF analysis was carried out in accordance with a previous study [38]. In brief, all sections were subjected to pre-chilled acetone fixation, 1 h incubation in blocking buffer (PBS based with 0.1% Triton X-100 and 5% normal donkey serum) at ambient temperature and overnight primary antibody incubation at 4 °C. On day 2, each section was rinsed three times with PBS before being incubated with secondary antibodies at room temperature in the dark. After 1 h, each section was mounted with DAPI (Abcam, Cambridge, UK) and staining was examined using a BX53-DP80 IF microscope (Olympus, Tokyo, Japan). The primary and secondary antibodies used in this study were as follows: rat monoclonal anti-Gr-1 (Cat. 14593182, 1:200, Invitrogen, Carlsbad, CA, USA), rat monoclonal anti-F4/80 (Cat. ab16911, 1:200, Abcam, Cambridge, UK), rabbit polyclonal anti-Ki67 (Cat. ab15580, 1:200, Abcam, Cambridge, UK), goat anti-rat IgG/FITC (Cat. SF132, 1:200, Solarbio, Beijing, China) and DyLight594 goat anti-rabbit IgG (H + L) (Cat. GAR007, 1:200, Multi Sciences, Hangzhou, China).

### 2.5. Quantitative Reverse Transcription Polymerase Chain Reaction (RT-PCR) Assay

Total RNA was isolated from primary human keratinocytes and skin tissues using an RNAeasy kit (Cat. 74104, Qiagen, Germantown, MD, USA) after which the total RNA was reverse transcribed into cDNA using a superscript III first strand kit (Cat. 1708891, Bio-Rad, Hercules, CA, USA). Takara SYBRR Premix Ex TaqTM II (Cat. RR820B, Takara Bio Inc., Mountain View, CA, USA) was used in the PCR procedure with the LightCyclerR 480 II (Roche Diagnostics, Risch-Rotkreuz, Switzerland). Following that, qRT-PCR amplification was performed in a 20 µL volume with 100 ng cDNA. The PCR procedure was as follows: 30 s at 95 °C; 40 cycles of 5 s at 95 °C, 20 s at 60 °C and 15 s elongation at 72 °C. Furthermore, Ct-values were calculated to quantify mRNA expression levels by 2^−ΔΔCt^, with 36β4 serving as the endogenous reference. The oligo sequences for genes used in PCR are shown in Appendix A.

### 2.6. ELISA

The ELISA kits to measure secreted IL-6 (Cat. VAL102), IL-8 (Cat. VAL103) and TNF-α (Cat. VAL105) were purchased from R&D Systems (Valukine TM Elisa, Minneapolis, MN, USA). The detection procedures followed the manufacturer’s protocols.

### 2.7. Western Blot Analysis

Total cellular proteins were extracted using RIPA lysis buffer containing 1% phosphatase inhibitor and 1% PMSF (Cat. P0100, Solarbio, Beijing, China). A BCA protein assay kit (Cat. PC0020, Solarbio, Beijing, China) was also used in this study to determine protein content. Following extraction, 12% SDS-PAGE was used to separate the proteins, which were then transferred to PVDF membranes (Cat. IPVH00010, Merck Millipore Ltd., Darmstadt, Germany). Later, 5% milk containing TBST (TBS powder supplemented with 0.1% Tween 20; pH 7.5) was added to block membranes for 1 h, after which the primary antibodies were added and incubated overnight at 4 °C. Primary antibodies used included: rabbit polyclonal anti-JNK (Cat. 9252, 1:1000, Cell Signaling Technology, Danvers, MA, USA), rabbit monoclonal anti-phospho-JNK (Cat. 4668, 1:1000, Cell Signaling Technology, Danvers, MA, USA), rabbit monoclonal anti-p38-MAPK (Cat. 8690, 1:1000, Cell Signaling Technology, Danvers, MA, USA), rabbit monoclonal anti-phospho-p38-MAPK (Cat. 4511, 1:1000, Cell Signaling Technology, Danvers, MA, USA), rabbit monoclonal anti-NF-κB (Cat. 8242, 1:1000, Cell Signaling Technology, Danvers, MA, USA), rabbit monoclonal anti-phospho-NF-κB (Cat. 3033, 1:1000, Cell Signaling Technology, Danvers, MA, USA), rabbit monoclonal anti-Erk1/2 (Cat. 4695, 1:1000, Cell Signaling Technology, Danvers, MA, USA), rabbit monoclonal anti-phospho-Erk1/2 (Cat. 4370, 1:1000, Cell Signaling Technology, Danvers, MA, USA), rabbit monoclonal anti-GAPDH (Cat. 5174, 1:5000, Cell Signaling Technology, Danvers, MA, USA), rabbit monoclonal anti-AMPKα (Cat. 5832, 1:1000, Cell Signaling Technology, Danvers, MA, USA), rabbit monoclonal anti-phospho-AMPKα (Cat. 2535, 1:1000, Cell Signaling Technology, Danvers, MA, USA), rabbit monoclonal anti-c-Myc (Cat. ab32072, 1:1000, Abcam, Cambridge, UK), rabbit monoclonal anti-mTOR (Cat. 2983, 1:1000, Cell Signaling Technology, Danvers, MA, USA), rabbit monoclonal anti-phospho-mTOR (Cat. 5536, 1:1000, Cell Signaling Technology, Danvers, MA, USA), rabbit monoclonal anti-p70S6K (Cat. 2708, 1:1000, Cell Signaling Technology, Danvers, MA, USA), rabbit polyclonal anti-phospho-p70S6K (Cat. 9205, 1:1000, Cell Signaling Technology, Danvers, MA, USA), rabbit monoclonal anti-4E-BP1 (Cat. 9644, 1:1000, Cell Signaling Technology, Danvers, MA, USA) and rabbit polyclonal anti-phospho-4E-BP1 (Cat. 9455, 1:1000, Cell Signaling Technology, Danvers, MA, USA). After rinsing three times with TBST for 5 min each, membranes were treated with a goat anti-rabbit secondary antibody (Cat. 7074, 1:5000, Cell Signaling Technology, Danvers, MA, USA) for 1 h at ambient temperature. In addition, an ECL chemiluminescence system (Cat. SW2010, Solarbio, Beijing, China) was used to observe protein bands and ImageJ software was used for analysis.

### 2.8. RNA-Seq Analysis

Total RNAs from keratinocytes treated with or without 1 mM phenformin for 10 h were isolated with Trizol reagent for RNA-seq analysis at Beijing Genomics Institute (BGI, Shenzhen, China) using the BGISEQ-500 system after transcription. The threshold value of RNA integrity number (RIN) was set to 7.0. Single-ended unstranded data bulk RNA-seq data were developed with a 50 bp read length and approximately 100–500 bp library size. STAR 2.5.3 was used to align fastq files against the GRCh38 reference genome. A gene count matrix was generated by the quantMode option to quantify reads per gene while mapping. DESeq2 (v1.26.0) was used to perform differential expression analysis in an R (v3.6.1) environment. Briefly, count matrices and sample information were imported into DESeqDataSet object; genes with less than 50 counts across all samples were discarded. Data were rlog transformed and sample distances were computed for downstream principal components analysis. Differential expression analysis was performed on raw count data by calling DESeq pipeline. A generalized linear model was generated to estimate the log fold changes in genes of interest. The *p* values of DEGs (differentially expressed genes) were corrected by implementing Benjamini–Hochberg (BH) testing and DEGs were extracted by setting the FDR cutoff smaller than 0.05 for downstream analysis, then subjected to volcano plots, circle graphs, gene clustering and hallmark pathway enrichment. The RNA-seq data were uploaded to the SRA database (NIH Sequence Read Archive) database (accession number PRJNA826690).

### 2.9. siRNA Transfection

siRNAs were transfected into keratinocytes as previously described [39]. In short, keratinocytes were seeded in 6-well plates. After reaching a density of 70%, lipofectamine 3000 was used to transfect various siRNAs into cells (Invitrogen, Carlsbad, CA, USA). The cells were harvested 72 h later to conduct RT-PCR and WB assays to determine silencing efficiency. In Gene Pharma, siRNA oligos targeting c-Myc and NC scrambled siRNA were obtained (Shanghai, China). To assess silencing efficiency, three different siRNAs were transfected for each gene. The oligo sequences of siRNAs used are shown in Appendix A.

### 2.10. Virus Infection

Keratinocytes (4 × 10^5^/well) were seeded in 6-well plates and incubated for 24 h. Following that, the c-Myc-expressing lentivirus or the relevant virus was transfected into cells for 6 h, followed by culture in growth medium for the time periods specified in the Figure Legends. Viruses were prepared and infected as previously described [39].

### 2.11. Statistical Analysis

GraphPad Prism 7 was used for statistical analysis (Graph Pad Software Inc., La Jolla, CA, USA). Means ± SD were used to represent the results. All assays were performed three times. This study used the Student’s *t*-test to compare two groups, while one-way or two-way ANOVA was used to compare multiple groups. A *p*-value less than 0.05 is considered to be statistically significant, which is indicated with “∗” in the corresponding Figures.

### 2.12. Ethics Statement

The Ethics Committee of Stomatological Hospital Shandong University approved each animal study (Protocol No. GR201720, Date: 27 February 2017). Each animal procedure was carried out in accordance with the National Institutes of Health’s Guidelines for the Care and Use of Laboratory Animals. The Partners Human Research Committee/IRB and Medical Ethical Committee of Shandong University’s School of Stomatology (Protocol No. GR201711, Date: 27 February 2017) approved the collection of human foreskin samples from hospital-discarded samples with no identifying information.

## 3. Results

### 3.1. Phenformin Suppresses Acute Inflammatory Responses Induced by TPA in the Skin

To test whether phenformin can play an anti-inflammation role, the pro-inflammatory compound TPA was applied on the ears of mice that had been systemically pretreated with either phenformin (Phen, 150 mg/kg) or PBS as a control. The thickness of each ear was measured at different time points and swollen and reddish ears were observed on the TPA-treated side, which was most significant at 8 h after TPA treatment, but not on the acetone plus PBS (Acet + PBS, a negative control) or the Acet + Phen or the TPA + Phen treated side (Figure 1A). The ear thickness increased significantly after TPA treatment compared to the control groups (Acet + PBS or Acet + Phen) and the increased ear thickness caused by TPA was suppressed by pretreatment with phenformin (TPA + Phen) (Figure 1B). Interestingly, the pretreatment with metformin (250 mg/kg) did not significantly reduce the increased ear thickness caused by TPA treatment (Appendix A). Histological analysis (HE stain) confirmed the increased ear thickness caused by TPA treatment, which was significantly inhibited by the addition of phenformin (TPA + phen) (Figure 1C, Appendix A), and we also observed that phenformin treatment reduced the thickness of subcutaneous edema and the infiltration of inflammatory cells (Figure 1C). To further confirm that phenformin blocks the infiltration of inflammatory cells induced by TPA treatment in the skin, IF staining for Gr-1 (Ly6) to label granulocytes (neutrophils) and for F4/80 to label macrophages was performed. The IF staining results showed an increased infiltration of immune cells in the skin treated with TPA, which was suppressed by the addition of phenformin (TPA + Phen) (red arrows, Figure 1D–G, lower magnification images are shown in Appendix A). Moreover, IF staining of Ki67, a proliferation marker, revealed that the increased number of proliferative cells in the dermis caused by TPA was also reduced by phenformin treatment (red arrows, Figure 1H,I, lower magnification images are shown in Appendix A). We did not observe any significant difference between the Acet + PBS and Acet + Phen control groups (Figure 1G–I, Appendix A).

### 3.2. Phenformin Suppresses Pro-Inflammatory Cytokine Expression in Ex Vivo Cultured Skin and in In Vitro Cultured Keratinocytes Induced by Poly I:C or by TPA

The infiltration of inflammatory cells into the skin is considered to be mainly mediated by cytokines, so we next analyzed whether the suppression of acute skin inflammation by treatment with phenformin was associated with a decreased production of cytokines in the skin. First, an ex vivo organ culture system was used to test whether phenformin could inhibit the increased cytokine expression in human skin induced by Poly (I:C), a well-known synthetic analog of viral dsRNA that induces inflammatory responses (Figure 2A). Poly (I:C), with or without different concentrations of phenformin, as indicated in Figure 2B, was applied on the surface (epidermis) of the skin, and 24 h later, the skin was collected for analysis of cytokine expression by qRT-PCR. Many pro-inflammatory cytokines, including IL-1β (IL-1B), IL-6, IL-8 (CXCL8), IL-12β (IL-12B), IL-23β (IL-23B), CCL2, CXCL16, TNFα and TGF-β1 (TGF-B1), as indicated in Figure 2B, were analyzed, and the results clearly show that the expression of all pro-inflammatory factors tested was induced by Poly (I:C) treatment without the addition of phenformin (0 mM) compared to the control group (0). However, the induction of cytokine expression by Poly (I:C) was significantly reduced by the addition of phenformin in a dose–response manner.

The above organ culture experiment was performed to measure cytokine expression in the whole skin, which contains different types of cells, including keratinocytes, dermal fibroblasts, immune cells and endothelial cells. Keratinocytes are usually the first type of cell in the skin that are exposed to external stresses and keratinocytes are able to produce different cytokines that play crucial roles in regulating skin inflammatory responses [40]. Therefore, we next tested whether phenformin can directly affect cytokine expression by keratinocytes. Primary keratinocytes derived from human skin tissues were treated with Poly (I:C) or TPA, with or without different concentrations of phenformin, and cytokine gene expression at 24 h after treatment was analyzed by RT-PCR and cytokine (protein) levels were measured by ELISA (Figure 2C). RT-PCR analysis of cytokine expression by keratinocytes treated with Poly (I:C), with or without different concentrations of phenformin, is shown in Figure 2D, and reveals that the expression of all pro-inflammatory cytokines tested was induced by Poly (I:C) treatment and that their induced expression by Poly (I:C) was significantly inhibited by the addition of phenformin in a dose–response manner. Those RT-PCR results were further verified by ELISAs of cytokines IL-6, IL-8 and TNF-α protein levels in the conditioned media collected at 24 h after Poly (I:C) treatment, with or without phenformin (Figure 2E). Similar results were observed using TPA to induce cytokine expression in keratinocytes and phenformin significantly suppressed cytokine expression increased by TPA (Figure 2F). Finally, we tested whether phenformin could inhibit the basal expression of cytokines in keratinocytes without any inflammatory stimulation. We found that phenformin could significantly suppress cytokine expression in human primary keratinocytes in in vitro cultures (Figure 2G). Taken together, these data suggest that phenformin can suppress the expression of pro-inflammatory cytokines or chemokines in human skin, likely directly through keratinocytes.

### 3.3. Phenformin Intially Suppresses the Expression of Pro-Inflammatory Cytokines but Not Directly by Blocking Activation of the NF-κB or MAPK Pathways in Human Keratinocytes

It is well-known that the production of inflammatory modulators is mainly controlled by the nuclear factor-kappa B (NF-κB), the mitogen-activated protein kinase (MAPK), the c-Jun N-terminal Kinase (JNK) and other signaling pathways [41,42]. Therefore, we next investigated whether phenformin suppresses the expression of pro-inflammatory cytokines by inhibiting those pathways in human keratinocytes. First, primary keratinocytes were treated with Poly (I:C), with or without different concentrations of phenformin, and, 24 h later, the cells were collected for Western blot analysis of phosphorylated and total levels of JNK, p38, Erk1/2 and NF-κB (Figure 3A). Those results showed that Poly (I:C) alone or combined with phenformin treatment did not affect the activation of JNK at 24 h, but that Poly (I:C) treatment significantly induced the phosphorylated level (activation) of p38, which was not affected by the addition of phenformin. However, Poly (I:C) treatment did not significantly induce the activation of Erk1/2, without (0.0 mM) or with low levels (0.5 mM) of phenformin. Interestingly, the activation level of Erk was higher in the Poly (I:C)-treated group together with 1.0 or 1.5 mM phenformin (black asterisks, Figure 3B) and, importantly, 1.0 mM and 1.5 mM phenformin inhibited the Erk1/2 activity compared to the control PBS (0.0 mM) group (red asterisks, Figure 3B). Finally, Poly (I:C) treatment significantly increased the phosphorylated level of NF-κB and the activation of NF-κB was significantly suppressed by the addition of different concentrations of phenformin (red asterisks, Figure 3B). This result suggested that phenformin suppresses the expression of pro-inflammatory cytokines, potentially through the down-regulation of the Erk and NF-κB pathways.

We then wanted to determine when the down-regulation of the basic expression of the Erk and NF-κB pathways, without any inflammatory stimulation, occurred after phenformin treatment. Therefore, human keratinocytes were treated with or without 1.0 mM phenformin and were collected at different time points for Western blot analysis of Erk1/2 and NF-κB activation levels (Figure 3C). Those results showed that the phosphorylated level of AMPK was significantly increased at 2 h after phenformin treatment compared with the non-treated control group, indicating that the AMPK pathway was already activated by phenformin at that time point. The increased phosphorylated level of AMPK in phenformin-treated cells was maintained until 24 h. However, the phosphorylated levels of Erk1/2 and NF-κB were not clearly changed at 2 h after phenformin treatment. We observed decreased phosphorylated levels of Erk1/2 or NF-κB at 4 h and 10 h after treatment, respectively. Quantification of the phosphorylated levels of AMPK, Erk1/2 and NF-κB is shown in Figure 3D and those results suggest that phenformin can activate the AMPK pathway as early as 2 h after treatment but suppresses the NF-κB and MAPK-ERK pathways at least 4 h after treatment. Next, we asked when the expression of inflammatory cytokines in human keratinocytes would start to be suppressed by phenformin treatment. Keratinocytes treated with similar conditions described for Figure 3C were collected for RT-PCR analysis of pro-inflammatory cytokines, which revealed that the decreased expression of all pro-inflammatory cytokines tested started as early as 2 h after phenformin treatment (Figure 3E), which was earlier than the time when activation of the Erk1/2 and NF-κB pathways was down-regulated (Figure 3C,D). Moreover, we observed that the down-regulated expression of pro-inflammatory cytokines elicited by treatment with phenformin became more significant with increased time of treatment, especially after 10 h, which was the time when strong inhibition of Erk1/2 or NF-κB activation by phenformin was observed (Figure 3C,D). Taking these data together, we conclude that the suppression of pro-inflammatory cytokine expression by phenformin in keratinocytes at early time points (as early as 2 h) was likely not directly through down-regulation of the MAPK or NF-κB pathways, which could contribute to the suppression of cytokine expression by phenformin at later time points.

### 3.4. RNA-Seq Analysis Reveals That Phenformin Significantly Suppresses the c-Myc Pathway at Earlier Time Points

In order to identify the potential targets by which phenformin suppresses the expression of pro-inflammatory cytokines in keratinocytes, those cells were treated with or without 1 mM phenformin for 10 h without adding an inflammation activator, either Poly (I:C) or TPA, and then were collected for RNA-seq analysis. These differentially expressed genes (DEGs) between two groups were analyzed by a volcano plot (Figure 4A), and the circle graph analysis revealed that 237 genes were up-regulated and 360 genes were down-regulated by phenformin treatment (Figure 4B); gene cluster analysis of DEGs is shown in Figure 4C. Those analyses revealed that several inflammation-related genes, including CXCL1, CXCL2 and CXCL8 (IL-8), were among the down-regulated DEGs. Gene Set Enrichment Analysis (GSEA) of DEGs revealed the top 20 enriched Hallmark pathways, which included 12 up-regulated and 8 down-regulated pathways, according to the normalized enrichment score (Appendix A). As expected, the well-known AMPK downstream targeted pathways, including glycolysis, metabolism, cell polarity (apical surface/apical junction) and p53, were among the list of up-regulated pathways (Appendix A). The inflammatory response and interferon response pathways appeared in the list of negatively regulated pathways (Appendix A). Interestingly, MYC-targets version 1 and 2 (V1/V2) were in the list of down-regulated pathways and were ranked the top two pathways according to the analysis of gene ratio, defined as the percentage of significant genes over the total genes in a given pathway (Appendix A and Figure 4D). The calculated enrichment scores further revealed that MYC-targets/inflammatory signaling, cytokine signaling as well as chemokine-signaling-related genes were mainly down-regulated in phenformin-treated cells (Figure 4E). The list of down-regulated MYC (also termed c-Myc) targets with phenformin treatment in RNA-seq data are shown in Appendix A and the expression of five well-known c-Myc target genes, CDK4, DDX19, APEX1, CAD and NOP16, together with c-Myc, were selected from Appendix A and were validated by RT-PCR analysis to be inhibited by phenformin treatment (Figure 4F). To further verify that phenformin suppresses c-Myc expression in human keratinocytes, we treated keratinocytes with different concentrations of phenformin and found that 0.5 mM phenformin significantly decreased the expression of c-Myc (Figure 4G). Next, we treated keratinocytes with 1 mM phenformin and collected cells for RT-PCR analysis of c-Myc. That analysis showed a decreased expression of c-Myc that appeared as early as 2 h after phenformin treatment (Figure 4H), which was further supported by the Western blot analysis of c-Myc protein expression in human keratinocytes treated with 1 mM phenformin (Figure 4I). Furthermore, both Poly (I:C) and TPA could induce c-Myc expression and that induction was down-regulated by phenformin treatment (Appendix A). Since phenformin is an AMPK activator, another well-known AMPK activator, AICAR was used to treat human keratinocytes and showed that AICAR also significantly suppressed c-Myc expression, as well as c-Myc targets (Figure 4J,K). Taken together, these data suggest that the AMPK activator phenformin suppresses c-Myc expression in human keratinocytes.

### 3.5. Inhibition of c-Myc Suppresses Cytokine Expression and the Over-Expression of c-Myc Rescues the Suppression of Cytokine Expression by Phenformin in Human Keratinocytes

As shown above (Figure 4H,I), the decreased expression of c-Myc appeared as early as 2 h after phenformin treatment, when the suppression of cytokine expression appeared (Figure 3). Considering that c-Myc has been reported to regulate cytokine expression, we hypothesized that the transcription factor c-Myc may be the target of phenformin treatment to suppress the expression of pro-inflammatory cytokines in keratinocytes. To test that hypothesis, we transfected human keratinocytes with a c-Myc siRNA to knockdown its expression and the knockdown efficiency was validated by RT-PCR and by Western blot (Figure 5A). We found that following the knockdown of c-Myc, the expression of pro-inflammatory cytokines, including IL-1β, IL-6 and IL-8, was significantly decreased compared to the control group (Figure 5B). As an alternative approach, a c-Myc inhibitor, JQ1, was used to further validate that the inhibition of c-Myc expression could down-regulate the expression of inflammatory cytokines. A similar result was found that the treatment with JQ1 significantly inhibited c-Myc expression (mRNA and protein, Figure 5C), as well as the expression of inflammatory cytokines, including IL-1β, IL-6 and IL-8 (Figure 5C). Interestingly, the inhibition of c-Myc expression did not seem to significantly affect TNFα expression in keratinocytes, suggesting that the regulation of TNFα may not be through the c-Myc pathway. Next, we increased the expression of c-Myc by stably expressing c-Myc in keratinocytes (Figure 5D), which showed that the overexpression of c-Myc induced the expression of pro-inflammatory cytokines (Figure 5E) and, importantly, that the overexpression of c-Myc could counteract the down-regulation of cytokine expression induced by treatment with phenformin (Figure 5F).

Next, we performed an in vivo study to test whether the inhibition of c-Myc could suppress the inflammation induced by TPA treatment. The ears of mice were treated with TPA alone or with TPA and the c-Myc inhibitor JQ1, or with acetone as a negative control. Swollen and reddish ears were observed on the TPA-treated side, but not on the acetone-treated or TPA + JQ1-treated side (Appendix A). The ear thicknesses at different time points were measured (Figure 5G) and showed that JQ1 treatment significantly reduced the ear swelling induced by TPA. Histological analysis (HE stain) confirmed that treatment with JQ1 reduced the ear thickness and the infiltration of immune cells (Figure 5H). The reduced infiltration of immune cells in JQ1-treated ears was verified by immunostaining of the immune cell markers Gr-1 and F4/80 (Appendix A and Figure 5I,J).

Taken together, these data suggest that phenformin targets c-Myc to suppress the expression of pro-inflammatory cytokines in keratinocytes and to inhibit inflammation in the skin.

### 3.6. Phenformin Controls c-Myc Expression through Inhibition of the mTOR Pathway in Keratinocytes

It has been shown that c-Myc expression is down-regulated by the AMPK/mTOR pathway, both in pancreatic tumor cells and in leukemia cells; therefore, we investigated whether phenformin also controls c-Myc expression through the mTOR pathway in keratinocytes. First, we treated keratinocytes with or without 1 mM phenformin and collected cells at different time points to check activation of the mTOR pathway. We found that phenformin could suppress the activation of the mTOR pathway, indicated by the reduced phosphorylated level of mTOR, as well as its downstream targets p70S6K and 4E-BP1, as early as 2 h after treatment, which was associated with the decreased expression of c-Myc in keratinocytes (Figure 6A,B). Then, we treated keratinocytes with the mTOR activator MHY1485 together with phenformin and found that activation of the mTOR pathway by MHY1485 resulted in an increased expression of c-Myc, which counteracted the phenformin-suppressing effect in keratinocytes (Figure 6C,D). Importantly, the mTOR activator MHY1485 enhanced the expression of cytokines, such as IL-6 and IL-8, to rescue the suppression of cytokine expression by phenformin (Figure 6E). The sum of these results indicated that phenformin down-regulates cytokine expression by targeting c-Myc through down-regulation of the mTOR pathway.

## 4. Discussion

It is well known that metformin, the most commonly prescribed drug for type II diabetes, plays antitumor functions in a large variety of tumors. It has been discovered that the effect of phenformin is more potent than metformin due to the way it enters cells because metformin is a hydrophobic compound and needs organic cation transporters (OCTs) to pass through the cellular membrane [43]. In contrast, phenformin does not require any transport(s) to freely enter cells [44]. Therefore, recent studies have demonstrated that phenformin is a promising anticancer agent with distinct dosing and usage, which has attracted the interest of many researchers [25,45,46]. Here, we found that phenformin, but not metformin, can efficiently suppress the inflammatory response induced by TPA in the skin. Inflammatory responses are mainly mediated by cytokines that are expressed by different types of cells, including immune cells, dermal fibroblasts and keratinocytes in the skin. Keratinocytes are the first type of cell in the skin that is exposed to extracellular stimuli due to its location at the most outermost layer of the skin, and keratinocytes are expected to play a major role in the initiation of inflammatory responses [47]. We found that phenformin can significantly inhibit the expression of pro-inflammatory cytokines in keratinocytes, which suggests that topical treatment with phenformin could suppress skin inflammation, likely through the regulation of keratinocyte cytokine expression. However, we also expected that phenformin would affect cytokine expression by dermal cells or immune cells to suppress skin inflammation, since we observed that the efficient suppression of cytokine expression in the skin (ex vivo) is higher than that of keratinocytes alone (in vitro).

Cytokines have been considered to be key modulators of both acute and chronic inflammation through the interactions of a complex network [48,49]. Our data demonstrated that phenformin can inhibit the expression of key pro-inflammatory cytokines, including IL-1β, IL-6 and TNFα, all of which signal via type I cytokine receptors, while the critical pro-inflammatory chemokine, IL-8, signals via G protein-coupled receptors (GPCRs) to suppress the skin inflammation response [50,51]. To understand how phenformin controls the expression of pro-inflammatory cytokines in keratinocytes, we initially investigated the MAPK and NF-κB pathways, two well-known signal transduction pathways involved in controlling the expression of proinflammatory cytokines [8,41,42,52]. The activation of p38, c-Jun N-terminal (JNK) kinase and extra-cellular-regulated kinases (Erk), the three major MAPK pathways, were investigated in keratinocytes treated with Poly (I:C), with or without phenformin. It has been previously shown that Poly (I:C) induces cytokine expression by the activation of MAPK pathways in keratinocytes; however, the activation status of JNK, Erk1/2 and p38 was different [53,54]. For instance, Mori et al. reported that Poly (I:C) significantly increased the activity of JNK, but not Erk and p38 at 1 h and 2 h treatment [53], but Dai et al. reported that Poly (I:C) significantly increased the activity of p38 as early as after 15 min of treatment, but not that of p38 or JNK [54]. Here, we found that at 24 h after treatment, Poly (I:C) did not significantly affect the activation of JNK or Erk, but significantly induced p38 activation. These differences of individual MAPKs are probably due to the observation time points. Interestingly, we found that 1 mM and 1.5 mM phenformin suppressed only Erk1/2 activity, but not the activity of JNK or p38. Poly (I:C) could significantly induce NF-κB activity, which is inhibited by phenformin. These data suggested that suppressing cytokine expression by phenformin might be through regulation of the Erk1/2 and/or NF-κB pathways. In order to make the system simpler, we treated keratinocytes with phenformin only and investigated when the inhibition of Erk1/2 and NF-κB occurred after treatment. We found that inhibition of the Erk1/2 or NF-κB pathways by phenformin started around 10 h later, but we observed that the suppression of cytokine expression by keratinocytes occurred as early as at 2 h after treatment with phenformin. This suggested that phenformin inhibits cytokine expression in keratinocytes, likely not through regulation of the MAPK or NF-κB pathways at early time points.

In order to characterize the mechanism by which phenformin initially controls cytokine expression in keratinocytes, we performed RNA sequence analysis of keratinocytes treated with or without phenformin, which revealed that the MYC pathway was significantly down-regulated in phenformin-treated keratinocytes. Notably, we observed a decreased expression of c-Myc in phenformin-treated keratinocytes as early as 2 h after treatment, a similar time point to the down-regulation of cytokine expression. Myc is a member of a family of transcription factors containing three related human genes: c-Myc (MYC), l-Myc (MYCL) and n-Myc (MYCN). C-Myc, usually referred to as MYC, was the first gene to be discovered in that family and has been shown to regulate a wide variety of biological processes, including division, apoptosis, cellular growth and angiogenesis, as well as inflammatory responses [55,56,57,58,59,60].

Inflammation has been shown to be involved in different stages of tumor development; c-Myc, a well-known oncogene, has been shown to activate inflammatory pathways to drive the formation of pro-tumoral inflammation [61,62]. C-Myc has been shown to act as a transcription factor that induces the transcription of genes encoding cytokines, including IL-2, IL-13 and IL-17C, in cancer cells [63,64]. Myc activation in pancreatic β cells was reported to induce the expression and release of the proinflammatory cytokine IL-1β [65]. The induction of Myc expression was shown to be associated with the induction of TNF-α and IL-6 in primary human blood macrophages treated with mycobacteria [66]. In agreement with those previous studies, we showed, in this study, that the inhibition of c-Myc, either by siRNAs or by the inhibitor JQ1 in keratinocytes, significantly suppressed the expression of pro-inflammatory cytokines IL-1β, IL-6 and IL-8, and that topical application of JQ1, similar to phenformin, could suppress skin inflammation in vivo. In contrast, the overexpression of c-Myc induced the expression of these pro-inflammatory cytokines and, importantly, the overexpression of c-Myc could counteract the suppression of cytokine expression induced by phenformin. Therefore, we conclude that phenformin suppresses cytokine expression in keratinocytes, likely through the down-regulation of c-Myc expression.

MYC expression is highly regulated and its expression level is tightly controlled by various mechanisms involving transcription (initiation and elongation), mRNA stability, translation and post-translation (protein stability) [67,68,69]. The AMPK pathway has been reported to control c-Myc expression through inhibition of the mTOR pathway in some cancer cells, including pancreatic neuroendocrine tumor cells, glioblastomas and leukemias [70,71,72,73]. Therefore, we expected that phenformin, as an AMPK activator, would suppress c-Myc expression through the AMPK/mTOR-axis-dependent pathway. Indeed, we observed the down-regulation of mTOR activity in phenformin-treated keratinocytes and the mTOR activator MHY1485 could induce c-Myc expression, and could counteract the suppression of c-Myc expression by phenformin in keratinocytes. The mTOR pathway is believed to regulate the activity of eukaryotic initiation factors (eIFs), such as eIF4e, to control c-Myc translation [72], but interestingly, both the protein and mRNA levels of c-Myc were down-regulated in phenformin-treated keratinocytes. C-Myc protein has been shown to bind the enhancer region of the c-Myc gene to control its own expression [74,75] and, furthermore, many cytokines, including IL-6, IL-1β, TNFα and NF-κB, have been reported to control c-Myc expression [62]. Therefore, it is not surprising that a decreased mRNA expression level of c-Myc was observed in phenformin-treated keratinocytes, but the detailed mechanism by which phenformin transcriptionally down-regulates c-Myc needs to be investigated in the future.

Skin inflammation is regulated by the crosstalk among keratinocytes with other skin cells, such as dermal fibroblasts or immune cells, or endothelial cells in the dermis. Therefore, it will be interesting to investigate whether the role of phenformin controls pro-inflammatory cytokine expression in other types of cells in the skin besides keratinocytes. We performed some preliminary studies, which showed that phenformin also decreases pro-inflammatory cytokine expression in dermal fibroblasts (Appendix A), which suggests that phenformin could play an anti-skin inflammation role through the regulation of different cells in the skin. A detailed study still needs to be performed to characterize this. Moreover, phenformin has been reported to play an anti-tumor function in several types of tumor cells and future work will be carried out to determine whether the anti-tumor role of phenformin is through the down-regulation of c-Myc together with the inhibition of inflammation in the skin.

In summary, the present study demonstrated that phenformin, an AMPK activator, suppresses the expression of pro-inflammatory cytokines, initially by downregulating c-Myc expression, followed by blocking the MAPK and NF-κB pathways to further decrease the expression of inflammatory cytokines in keratinocytes, which suggests that phenformin can be developed into a potential therapeutic drug for the treatment of skin inflammation in the future.

## Figures and Tables

**Figure 1 cells-11-02429-f001:**
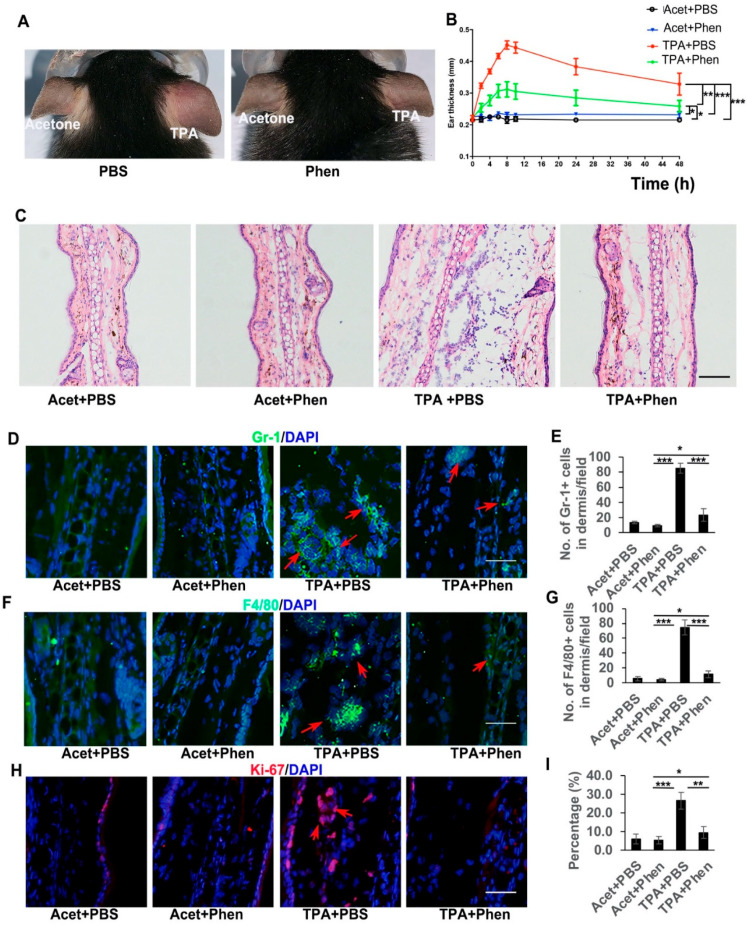
Phenformin suppresses acute skin inflammation induced by TPA. (**A**) Representative images of the ears of mice treated with PBS (as a control) or with phenformin (Phen, 150 mg/kg) with the topical application of acetone on the left ears or TPA on the right ears at 8 h after treatment. (**B**) Ear thicknesses of mice from (**A**) were measured at different time points as indicated. Two-way ANOVA with correction for multiple pairwise comparisons was used for statistical analysis of differences between two groups as indicated, *n* = 3, standard deviation bars are shown, * *p* < 0.05; ** *p* < 0.01; *** *p* < 0.005. (**C**) Histological analysis of the ears of mice from (**A**) at 8 h after TPA treatment, quantification of ear thickness seen by HE stain is shown in Appendix A; (**D**–**I**) Sections from (**C**) were analyzed for infiltration of inflammatory cells by IF analysis of Gr-1 ((**D**), green) and F4/80 ((**F**), green) and cell proliferation by IF analysis of Ki67 ((**H**), red); DAPI is used as a counterstain for nuclei, lower-magnification images of (**D**,**F**,**H**) are shown in Appendix A. The corresponding quantification analysis of the number of Gr-1 or F4/80 or Ki67-positive cells per field (**D**,**F**,**H**) is shown in (**E**) (Gr-1), (**G**) (F4/80) and (**I**) (Ki67). (**E**,**G**,**I**): Student’s *t* test analysis was used for all quantification data to compare two groups as indicated, *n* = 3, standard deviation bars are shown, * *p* < 0.05; ** *p* < 0.01; *** *p* < 0.005. Bars: 100 μm in (**C**); 50 μm in (**D**,**F**,**H**).

**Figure 2 cells-11-02429-f002:**
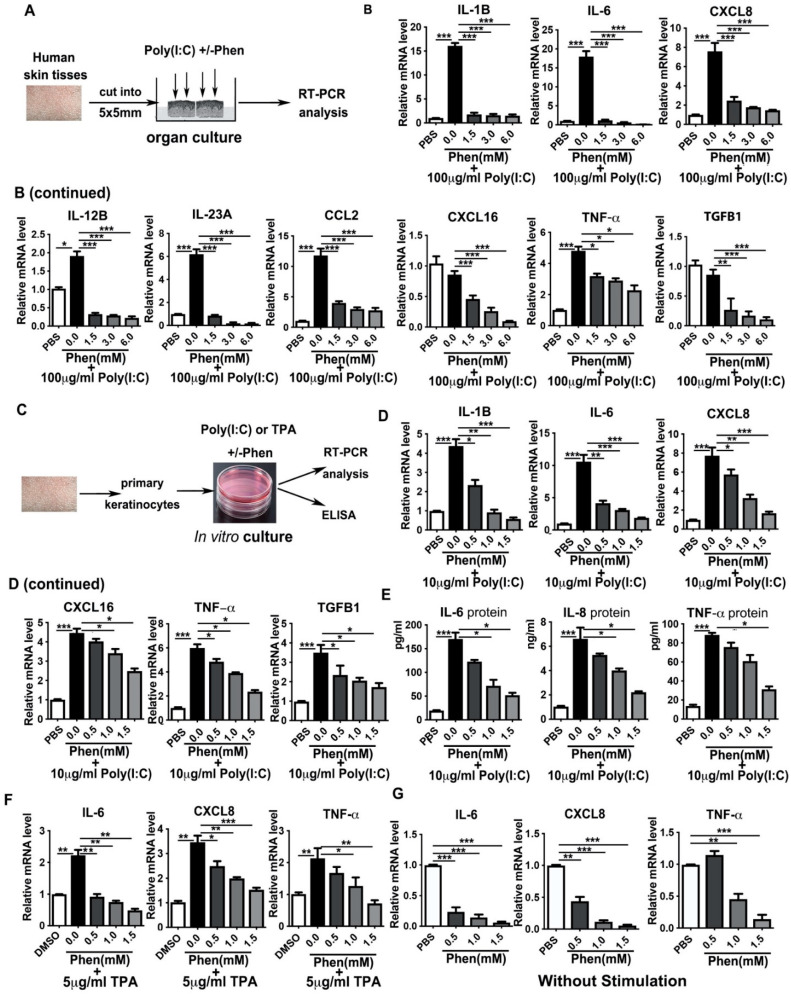
Phenformin suppresses the expression of pro-inflammatory cytokines in human keratinocytes. (**A**) Scheme of human skin organ culture: skin tissues were treated with 100 μg/mL Poly (I:C) with or without different concentrations (0, 1.5, 3.0 or 6.0 mM) of phenformin (Phen) or PBS as a negative control for 24 h, then were collected for RT-PCR analysis of cytokine expression. (**B**) Total mRNAs were extracted from skin tissues from (**A**) and qRT-PCR was used to analyze the expression of the indicated pro-inflammatory cytokines; results were normalized to levels of 36B4 mRNA. (**C**) Scheme of in vitro culture of primary human keratinocytes treated with 10 μg/mL Poly (I:C) or with 5 μg/mL TPA with or without different concentrations (0, 0.5, 1.0 or 1.5 mM) of phenformin (Phen) or PBS as negative control for 24 h, after which the cells were collected for RT-PCR analysis of cytokine expression. (**D**) Total mRNAs were extracted from keratinocytes treated with Poly (I:C) combined with phenformin and qRT-PCR analysis of expression of the indicated pro-inflammatory cytokines; results were normalized to levels of 36B4 mRNA. (**E**) ELISA analysis of the indicated cytokine protein levels in the conditioned medium collected from keratinocytes treated with Poly (I:C) combined with phenformin as in (**C**). (**F**) Total mRNAs were extracted from keratinocytes treated with TPA combined with phenformin and qRT-PCR analysis of the expression of the indicated pro-inflammatory cytokines; results were normalized to levels of 36B4 mRNA. (**G**) Total mRNAs were extracted from keratinocytes treated with PBS or with different concentrations (0, 0.5, 1.0 or 1.5 mM) of phenformin and qRT-PCR analysis of the expression levels of the indicated pro-inflammatory cytokines; results were normalized to levels of 36B4 mRNA. (**B**,**D**–**G**): Student’s *t* test analysis was used for all quantification data to compare each different concentration group as indicated with the 0.0 concentration of phenformin, *n* = 3, standard deviation bars are shown, * *p* < 0.05; ** *p* < 0.01; *** *p* < 0.005.

**Figure 3 cells-11-02429-f003:**
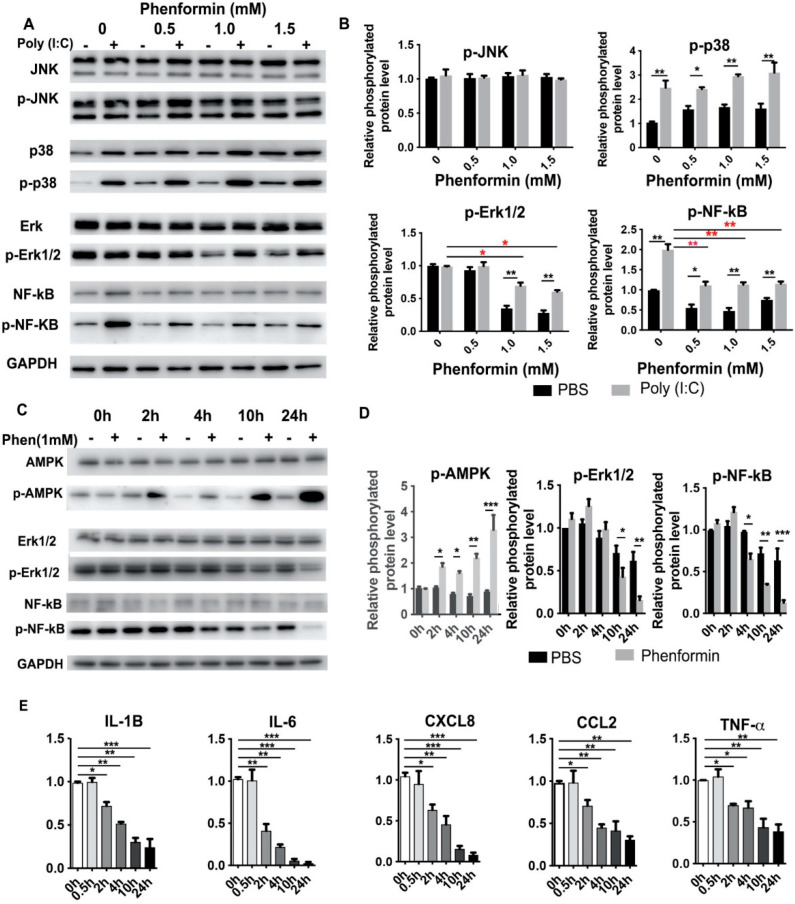
Phenformin suppresses the expression of pro-inflammatory cytokines in human keratinocytes but not directly through inhibition of the NF-κB or MAPK signaling pathways at early time points. (**A**,**B**) Primary human keratinocytes were treated with 10 μg/mL poly (I:C) with (+) or without (−) different concentrations (0, 0.5, 1.0 or 1.5 mM) of phenformin as indicated. At 24 h after treatment, cells were collected for Western blot analysis of total and phosphorylated levels of JNK, p38, Erk1/2 and NF-κB proteins; GAPDH was used as a loading control. The levels of the phosphorylated forms of these proteins relative to the control PBS-treated cells (expression level as 1) are shown in (**B**), and densitometry measurements for phosphorylated forms were normalized to the amounts of total proteins, respectively. (**C**,**D**) Primary human keratinocytes were treated with 1.0 mM phenformin (Phen) (+) or with PBS (−) and cells were collected at different time points for Western blot analysis of total and phosphorylated levels of AMPK, Erk1/2 and NF-κB proteins; GAPDH was used as a loading control. Levels of the phosphorylated forms of these proteins relative to the control PBS-treated cells (expression level as 1) are shown in (**C**), and densitometry measurements for the phosphorylated forms were normalized to the amounts of total proteins. (**E**) Primary human keratinocytes were treated with the same conditions as described for C and cells were collected at different time points as indicated for qRT-PCR analysis of the expression of the indicated pro-inflammatory cytokines; all mRNA expression levels were normalized to levels of 36B4 mRNA and are shown as relative to the control group (PBS treated, as 1). (**B**,**D**,**E**): Student’s *t* test analysis was used for all quantification data to compare two different concentration groups as indicated, *n* = 3, standard deviation bars are shown, * *p* < 0.05; ** *p* < 0.01; *** *p* < 0.005.

**Figure 4 cells-11-02429-f004:**
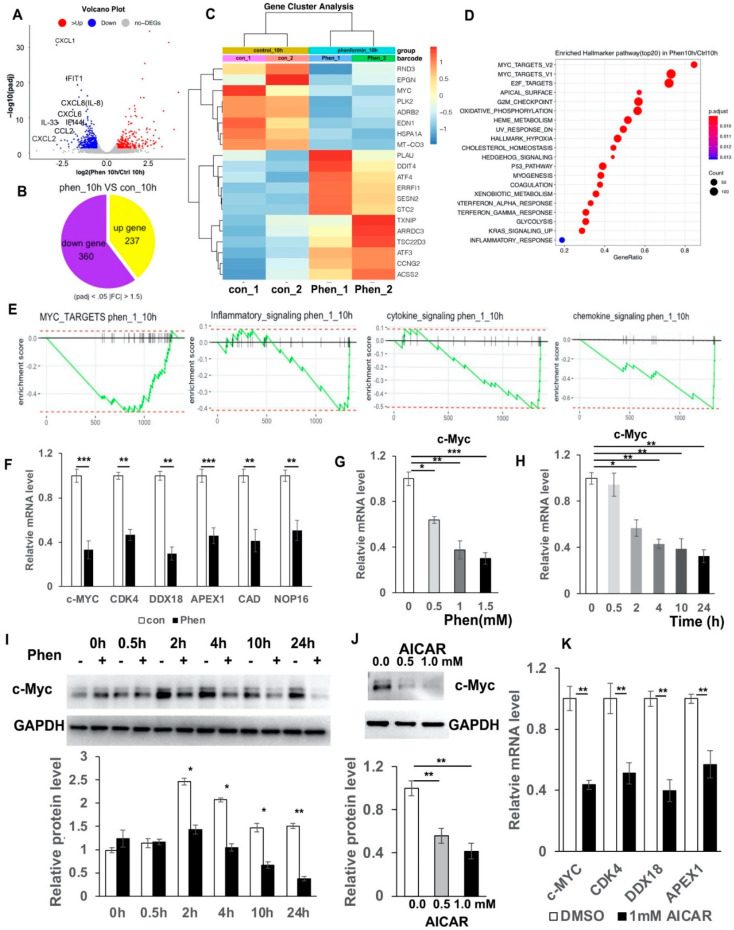
RNA-seq analysis reveals that phenformin suppresses c-Myc expression in human keratinocytes. (**A**) Volcano plot visualizing DEGs between phenformin (1 mM)-treated cells (phenformin_10h) and the control PBS-treated cells at 10 h (control_10h); a *p* value < 0.001 was used as a threshold to determine the significance of DEGs. Red dots represent up-regulated DEGs, blue dots represent down-regulated DEGs and gray dots indicate transcripts that did not change significantly between the two groups Some pro-inflammatory cytokines from down-regulated DEGs are indicated. (**B**) Circle graph presenting the number of DEGs between the phenformin-treated group and the control group at 10 h. (**C**) Gene cluster analysis of DEGs between phenformin-treated (Phen_1, Phen_2) and control cells (con_1, con_2) was conducted based on the FPKM value of each sample. The *X* axis represents the different samples, whereas the *Y* axis represents DEGs. The color (from blue to red) represents DEG expression intensities from low to high, respectively. (**D**) Top 20 enriched Hallmarker pathway of DEGs ranked by gene ratio, the *X* axis shows the gene ratio, which is defined as the percentage of significant genes over the total genes in a given pathway; the left *Y* axis shows the top 20 pathway names. The darker the color the smaller the q value; bubble size indicates the number of DEGs. (**E**) Enrichment plots for 4 selected datasets enriched in GSEA Hallmark analysis from (**D**), showing the profile of the running ES Score and positions of gene set members on the rank-ordered list. (**F**) Relative mRNA expression levels of c-Myc and its downstream targets CDK4, DDX18, APEX1, CAD and NOP16, normalized by the human 36β4 gene, in human keratinocytes treated with control PBS (con) or with 1 mM phenformin at 10 h by RT-PCR analysis. (**G**,**H**). Relative mRNA expression levels of c-Myc normalized by the human 36β4 gene, in human keratinocytes treated with different concentrations (0, 0.5, 1.0 or 1.5 mM) phenformin (Phen) at 10 h (**G**); or treated with 1 mM phenformin and collected at different time points (0, 0.5, 2, 4, 10 and 24 h) (**H**), by RT-PCR analysis. (**I**) Western blot analysis of c-Myc protein levels in human keratinocytes treated with 1 mM phenformin and collected at different time points (0, 0.5, 2, 4, 10 and 24 h). Densitometry measurements of the relative expression level of c-Myc normalized to the loading control GAPDH are shown in the lower panel. (**J**) Western blot analysis of c-Myc protein levels in human keratinocytes treated with 0.5 or 1.0 mM AICAR for 24 h. Densitometry measurements for the relative expression level of c-Myc normalized to GAPDH as a loading control are shown in the lower panel. (**K**) Relative mRNA expression levels of the indicated genes normalized by the human 36β4 gene, in human keratinocytes treated with 1.0 mM AICAR for 12 h. Student’s *t* test analysis was used for all quantification data to compare two different concentration groups as indicated, *n* = 3, standard deviation bars are shown, * *p* < 0.05; ** *p* < 0.01; *** *p* < 0.005.

**Figure 5 cells-11-02429-f005:**
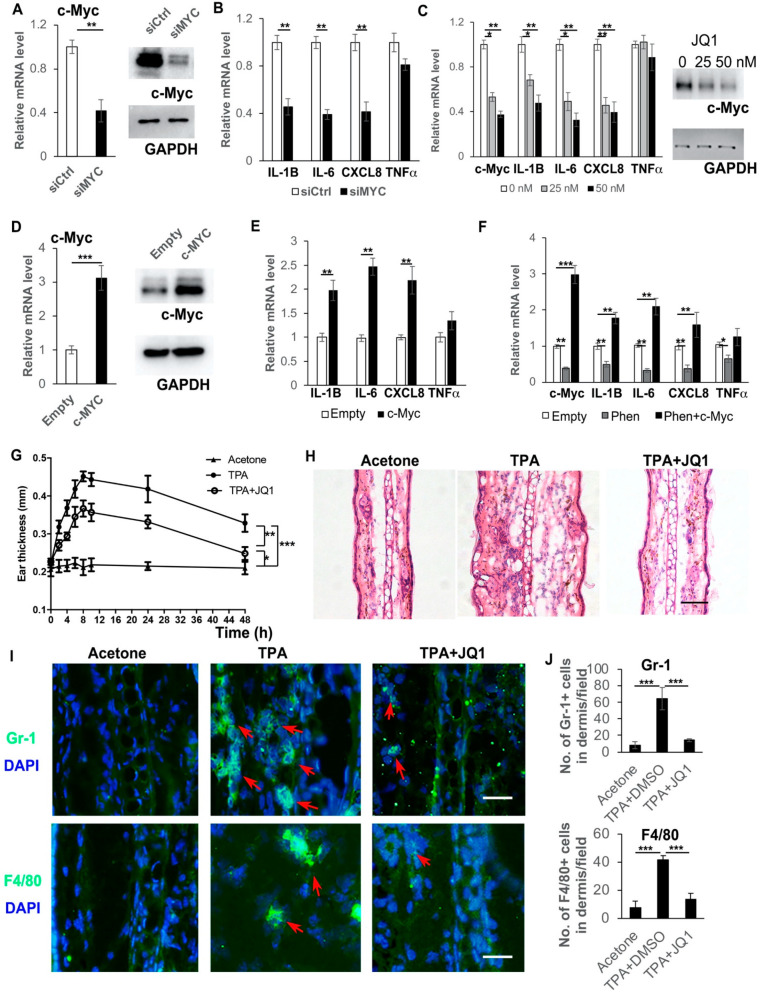
c-Myc controls the expression of pro-inflammatory cytokines in human keratinocytes. (**A**,**B**) Primary human keratinocytes were transfected with an siRNA for c-Myc (siMYC) or a control scramble siRNA (siCtrl) and 72 h after transfection, cells were collected for RT-PCR analysis of c-Myc mRNA expression (left panel, (**A**)), Western blot analysis of c-Myc protein expression (right panel, (**A**)) and RT-PCR analysis of mRNA expression levels of pro-inflammatory cytokines as indicated (**B**). (**C**) Primary human keratinocytes were treated with 0, 25 or 50 nM c-Myc inhibitor JQ1 and cells were collected at 24 h for RT-PCR analysis of mRNA expression levels of pro-inflammatory cytokines as indicated. (**D**) Primary human keratinocytes were infected with a c-Myc-expressing virus (c-Myc) or a control virus (empty), and 48 h after infection, cells were collected for RT-PCR analysis of c-Myc mRNA expression levels (left panel, (**D**)), Western blot analysis of c-Myc protein expression (right panel, (**D**)) and RT-PCR analysis of mRNA expression levels of pro-inflammatory cytokines as indicated (**E**). (**F**) Primary human keratinocytes were infected with a c-Myc-expressing virus (c-Myc) or a control virus (empty), and 12 h later, cells were treated with 1 mM phenformin for 48 h after infection and cells were collected for RT-PCR analysis of expression levels of c-Myc and pro-inflammatory cytokines as indicated. (**G**) Mouse ears were treated with following combinations: Left ears: Acetone (Negative control); Right ears: TPA (positive control) or TPA + JQ1. Ear thickness was measured at different time points as indicated. Two-way ANOVA with correction for multiple pairwise comparisons was used for statistical analysis of differences between two groups as indicated, *n* = 3, standard deviation bars are shown, * *p* < 0.05; ** *p* < 0.01; *** *p* < 0.005. (**H**) Histological analysis of ears of mice from (**G**), at 8 h after TPA treatment; (**I**,**J**) sections from (**H**) were analyzed for infiltration of inflammatory cells by IF analysis of Gr-1 ((**I**), green) and F4/80 ((**I**), green); DAPI was used as a counterstain for nuclei, red arrows indicate positive cells. Lower-magnification images of (I) are shown in Appendix A. The corresponding quantification analysis of ((**I**) and Appendix A) is shown in (**J**), Student’s *t* test analysis except (**G**) was used for all quantification data to compare two different concentration groups as indicated, *n* = 3, standard deviation bars are shown, * *p* < 0.05; ** *p* < 0.01; *** *p* < 0.005. Bars: 100 μm in (**H**); 50 μm in (**I**).

**Figure 6 cells-11-02429-f006:**
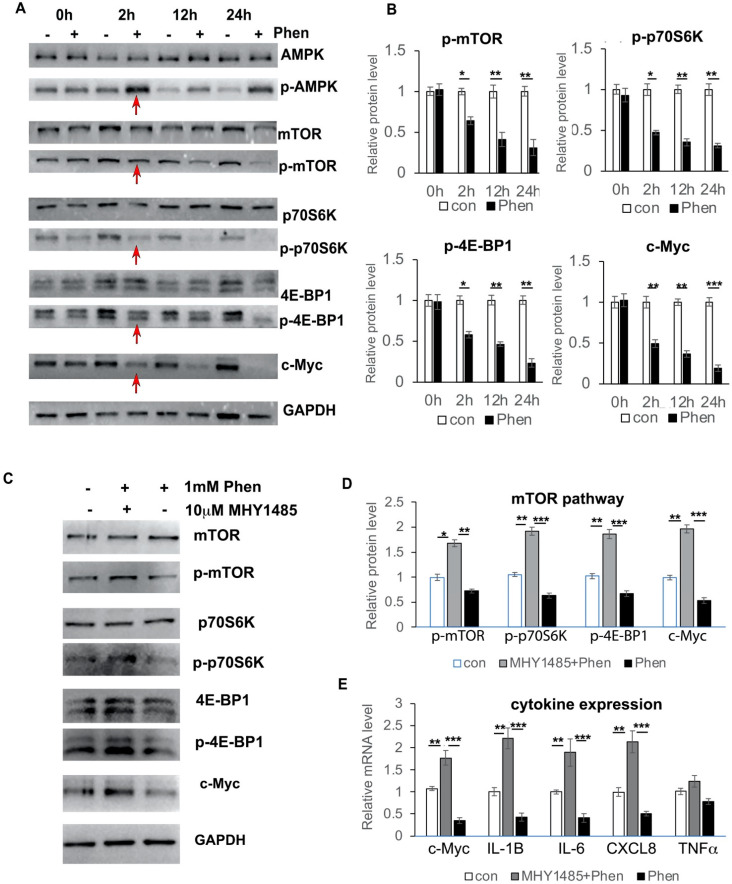
Phenformin controls c-Myc expression through down-regulation of the mTOR pathway in human keratinocytes. (**A**,**B**) Primary human keratinocytes were treated with 1 mM phenformin (+) or PBS (−) and were collected at different time points for Western blot analysis of total and phosphorylated levels of AMPK, mTOR, p70S6K and 4E-BP1 proteins, as well as c-Myc; GAPDH was used as a loading control. Levels of the phosphorylated forms of these proteins relative to the control PBS-treated cells (expression level as 1) are shown in (**B**); densitometry measurements for the phosphorylated forms were normalized to the amounts of total proteins and c-Myc was normalized by GAPDH. Red arrows indicate changes in these protein levels at 2 h. (**C**,**D**) Primary human keratinocytes were treated with 1 mM phenformin (Phen) with or without 10 μM mTOR activator MHY 1485 or PBS (−) and were collected at 2 h for Western blot analysis of total and phosphorylated levels of mTOR, p70S6K and 4E-BP1 proteins, as well as c-Myc; GAPDH was used as a loading control. Levels of the phosphorylated forms of these proteins relative to the control PBS-treated cells (expression level as 1) are shown in (**D**); densitometry measurements for the phosphorylated forms were normalized to the amounts of total proteins and c-Myc was normalized by GAPDH. (**E**) Primary human keratinocytes were treated with the same conditions as (**C**) and cells were collected at 2 h for RT-PCR analysis of mRNA expression levels of the indicated genes normalized by the human 36β4 gene. (**B**,**D**,**E**): Student’s *t* test analysis was used for all quantification data to compare two different concentration groups as indicated, *n* = 3, standard deviation bars are shown, * *p* < 0.05; ** *p* < 0.01; *** *p* < 0.005.

## Data Availability

The raw data in this study can be found in the following link: https://u.pcloud.link/publink/show?code=XZEoi8VZhiPgyGOhX5RJXC5AnDAMV7GkkGly (accessed on 1 August 2022).

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
