# Peer review of "Phenformin Down-Regulates c-Myc Expression to Suppress the Expression of Pro-Inflammatory Cytokines in Keratinocytes"

_cells, 2022, doi:10.3390/cells11152429_

Round 1

Reviewer 1 Report

In this manuscript, Liu et al. uncovers some details regarding the mechanism by which phenformin dampens pro-inflammatory gene expression in epidermal keratinocytes.

The paper is interesting, fairly thorough, and presents a good amount of data. I think it will be a valuable addition to the literature after addressing the numerous, but mostly minor, points enumerated below.

In the abstract, the authors should spell out the chemical TPA, since they have not yet defined the abbreviation.

In the introduction, the authors do a good job of describing in brief the role of keratinocytes in inflammatory skin diseases, as sensors and executors of inflammatory paradigms. It is probably also worth the authors addressing how aberrant epidermal programmes, like hyperproliferation and disrupted differentiation, contribute to lacking epidermal barrier function. This correspondingly leads to an increase in TEWL, further leading to inflammation of the epidermis, continuing the cycle. Indeed, reduced hydration appears to be both a symptom and a driver of exacerbated dermatitis in many cases.

Authors state “….no study has reported the role of metformin in human skin inflammation or in the expression levels of pro-inflammatory cytokines in human primary keratinocytes or in the skin.” This does not appear to be strictly true; see Tsuji et al. 2020 “ Metformin inhibits il-1B secretion via impairment of NLRP3 inflammasome in keratinocytes: implications for preventing the development of psoriasis.” There may be others as well but this was one of the first things that came up in a quick google scholar search. I concede that likely not many studies have examined this, but it does not seem to be accurate to say “ no study.”

Use of “70’s” is colloquial. The authors should instead state “the 1970s.”

In the methods section, the RNA-seq analysis should have more details regarding library prep, sequencing, etc. What kind of prep (mRNA, total RNA) was performed and using which reagents? Was there a RIN cutoff? Was the prep stranded or unstranded? What was the read length? What was the (approximate) library size? Was the sequencing single-end or paired end? How were fold change values calculated? Were FPKM values compared directly? Or were DEGs determined using EdgeR, DESeq2, etc.?

For statistical analysis, for groups compared using ANOVAs, were post-hoc comparisons used to evaluate pairwise statistical significances?

Authors should be careful for how they describe their antibodies in the methods section. The authors say “The following primary and secondary antibodies were 150 used: monoclonal anti-mouse Gr-1 (Cat. 14593182, 1:200, Invitrogen, Carlsbad, CA, United 151 States), monoclonal anti-rat F4/80 (Cat. ab16911, 1:200, Abcam, Cambridge, United Kingdom)”. In reality, according to the catalog entries, what the authors describe as “anti-mouse Gr-1” is an antibody made in rat that targets mouse Gr-1, but what the authors describe as “monoclonal anti-rat F4/80” is also an antibody made in rat that targets mouse F4/80. This was initially confusing to me because I wondered how authors got specific staining using an antibody (anti-Gr1) developed in mouse in order to stain specifically in mouse tissue, but then I examined the datasheet and saw that it was actually raised in rat. Authors should make sure the species of origin of these antibodies are clear in this section.

For figure 1H, authors show distinct differences in the amount of proliferating cells among the groups based on Ki67 staining. Are the quantifications of Ki67-positive cells inclusive of both dermal and epidermal cells? Or are authors only counting one or the other? The quantifications here are also described as “percentages” but it’s not clear what this means. E.g. if the number of 60%, 60% of which cells are positive?

Also, regarding Figure 1, I appreciate that the authors included relatively low power images to make it obvious that they are not biasing their counts; there are clear differences among groups for the markers. But the images being low power make it difficult to speculate as to what populations of cells are proliferative, which could yield interesting information for the reader, for Ki67, and make it difficult to evaluate the specificity of the staining for the other immune cell markers. I would appreciate if authors could either include an inset with higher power images (or magnifications of parts of the existing images including both epidermis and dermis) in this figure, or, perhaps easier, to include another supplementary figure with higher powered images, just to give a good idea of what kinds of cells (keratinocytes, immune cells, fibroblasts, endothelial cells, etc.) in the epidermis and dermis are proliferative and not proliferative, as well as to better assess the specificity of their immunofluorescent immune cell staining, as it is very difficult at low magnification.

Line 301 what was the Poly I:C dissolved in? Does it naturally penetrate through human stratum corneum to reach live keratinocytes in order to cause inflammation?

Did the experiments in figure 2 also involve pretreatment with phenformin, as in the animal experiments in fig 1? Or were they co-treatment with I:C and phenformin?

Some of the y-axis labels in figure 2 overlap with their titles (the numbers and text overlap)

Authors should consider using the gene names for the qRT-PCR results, and keeping the protein names for ELISA data.

In 3.2, authors should list the genes tested in the results section. I know there are a lot of them, but the way it is worded currently, a reader doesn’t know what any of the genes are until he or she looks at the figure.

Line 375-377 were these also treated with I:C or TPA? Or were these just looking at the effects of phenformin on basal levels of activation of ERK and NF-kB?

Line 394-397 I understand what the authors are saying, but their explanation may be slightly too simplistic. What the authors show is that decrease in pro-inflammatory gene expression precedes detectable decrease in MAPK or NF-kB activity. This suggests that the initial stimulus of phenformin leads to an initial decrease in inflammatory gene expression in a manner that is independent of ERK/NF-kB activity. However, several hours later, the authors do detect decreases in activation of these pathways, and these decreases are accompanied by even larger downregulations of inflammatory gene expression (for example, IL-1B is about 25% downregulated at 2 hours but more than 75% downregulated at 24 hours). Therefore, though the authors are apparently correct that the initial stimulus is independent of these pathways, the fact that decrease MAPK and NF-kB activity are detected at later timepoints likely represents a scenario where reduction of activities in these pathways may contribute to greater degrees of downregulation of inflammatory genes at later time points (e.g. >4hours). Thus, the summary in line 394-397 lacks nuance and should be restated. In particular, I do not think it is fair to group ERK/NF-kB (which are likely players at later time points) with JNK and p38 (which do not appear to be relevant at all regarding the inhibitory effects of phenformin on inflammatory gene expression) by saying all of them appear to be uninvolved.

Line 422-423: What concentration of phenformin was used for the RNA-seq analysis? Also, how long were cells treated? Please include these in the results section as well as the figure legend. Were these cells stimulated with TPA or I:C or just baseline unactivated?

Authors’ identification of CXCL1, CXCL2, and CXCL8 genes as all downregulated by phenformin is especially interesting, as I believe all of these genes encode neutrophil chemoattractants.

Authors discuss a few of the genes downregulated by phenformin, but I wonder if any of the other keratinocyte genes commonly associated with, for example, psoriasis are also found to be downregulated such as IL-17, S100A7, S100A8, S100A9, S100A12, antimicrobial peptides, etc.?

In the RNA-seq data, is there anything interesting about the genes or pathways that are upregulated by phenformin treatment? Authors really only discuss the downregulated genes (and use it to discover the role of Myc), but the volcano plot shows a large number of upregulated genes as well. Are any of these genes differentiation markers or barrier genes? Aquaporins, keratins, loricrin, filaggrin, involucrin, etc.?

Same comments about the figure 5 immunofluorescent images as I made about the figure 1 images.

Line 627 MAKP is a typo.

Line 624-627, same comment as above about making this explanation more nuanced.

Authors should include some future directions at the end of the discussion section. What are the important questions left unanswered, in the authors’ opinions? Personally, I would be interested in seeing more about how these compounds might affect gene expression and cellular phenotypes in isolated dermal cells. Do they act on immune cells or fibroblasts or endothelial cells in the dermis directly? Or are the effects on blunting dermal inflammation in vivo based solely on the lack of chemotactic signaling emanating from keratinocytes due to phenformin acting directly on them?

Reviewer 2 Report

Liu et al, in this manuscript demonstrated that Phenformin antidiabetic drug from biguanide drug class have potential to suppress the pro-inflammatory cytokines in keratinocytes via down regulation of c-Myc expression. This drug is currently not used in clinic. In this study authors used mouse model for acute inflammation, ex vivo human skin organ culture system and in vitro human primary keratinocytes to demonstrate these findings. In summary, first authors showed that in mouse model phenformin, significantly suppress the inflammatory response induced by Tetradecanoylphorbol-13-acetate (TPA) in the skin by measuring ear thickness, histological and immunostaining of skin tissue with and without with phenformin treatment. Next authors used ex vivo human skin organ culture and human primary keratinocytes culture induced by poly I:C or TPA to demonstrate the anti-proinflammatory effect of phenformin. Authors measured the expression of inflammatory cytokines by qRT-PCR. Further authors examined the regulators of inflammatory modulators and reported that the effect of suppression of pro-inflammatory cytokines expression by phenformin was not occurs via inflammation modulator signaling pathways such as JNK, p38, Erk1/2 and NF-kB. Next authors performed RNA seq analysis to identify potential targets of phenformin which suppresses the expression of proinflammatory cytokines in keratinocytes. RNA seq results showed that numerous inflammation-related genes such as CXCL1, CXCL2 and CXCL8 were among the downregulated genes. The calculated enrichment scores further revealed that MYC-targets/inflammatory signaling, cytokine signaling as well as chemokine signaling related genes were mainly downregulated in phenformin treated cells. The expression of 5 well-known c-Myc target genes (CDK4, DDX19, APEX1, CAD and NOP16) together with c-Myc were downregulated by phenformin treatment which was further validated by qRT-PCR. Next authors showed that the AMPK activator phenformin suppresses c-Myc expression and suppressed pro-inflammatory cytokines expression in human keratinocytes which was further validated by inhibiting and overexpressing c-Myc in keratinocytes. Finally, authors showed that phenformin downregulates cytokine expression by targeting c-Myc inhibition by downregulation of the mTOR pathway in keratinocytes.

This study provides a novel role of phenformin in inhibiting the pro-inflammatory response in keratinocytes by downregulating mTOR pathway which in turn downregulate the c-Myc expression. This study demonstrated that phenformin as a potential therapeutic option to treat skin inflammation.

This manuscript can be further improved by addressing the following concerns.

Major comments:

1. In this study authors mainly focus on keratinocytes it would better if authors also performed few experiments with skin fibroblast to confirm that that the effect of phenformin is mainly on keratinocytes and not with fibroblast or with both cell types.

2. In Figure 2B authors used 1.5 mM to 6.0 mM concentration of phenformin to treat human skin organ culture. Effect of phenformin is significant at 1.5mM concentration, it would be better if authors also used lower concentration 0.5 and 1.0 mM to demonstrate that this drug have lower, medium effect and higher effect in this culture system.

Minor comments:

1. In Figure 5C, authors showed different dose of JQ1 treatment to inhibit c-Myc expression and measure c-Myc expression at mRNA level only it would be better if authors also show c-Myc protein levels in this treatment conditions.

2. In Figure 2G, Y axis labeling in overlapping with Y axis numbers.

Reviewer 3 Report

In the manuscript “Phenformin downregulates c-Myc expression to suppress the expression of pro-inflammatory cytokines in keratinocytes”, Liu et al investigate the effects of phenformin on skin inflammation and explore the potential mechanisms. The authors found phenformin can suppress acute skin inflammatory responses through the downregulation of c-Myc expression with human skin tissue models. This work is interesting and has tremendous clinical potential. The experimental design is not rigorous, and the writing needs to be improved. There are several suggestions to improve this manuscript.

1. In the “2.1. Isolation and culture of primary human keratinocytes” section, the authors should briefly introduce the processes of how to isolate and culture the primary human keratinocytes in vitro.

2. The authors wrote “Since we didn’t observe any significant difference between the acetone+PBS and the Acetone+Phen treated ears, acetone+Phen was used as a control group for the IF staining.” The design is unreasonable, the Acetone + PBS group was required.

3. Several reports demonstrated that JNKs were activated by phosphorylation induced by poly(I:C) stimulation, such as PMID: 29395576. “JNKs were activated by phosphorylation induced by poly(I:C) stimulation [39], as we have reported previously in keratinocytes [36]”.  But the p-JNK was not activated after treatment with poly(I:C) in keratinocytes. The authors should discuss it in the discussion section.

4. In the results section, “3.3. Phenformin suppresses the expression of pro-inflammatory cytokines but not by blocking the activation of the NF-kB or MAPK pathways in human keratinocytes”, the subtitle is confusing, because the p-NF-kB activate is significantly suppressed by the addition of phenformin.

5. In figure 3C-E, the authors treated the human keratinocytes with Phen alone. The design makes puzzling. The previous work was focused on the skin inflammatory model induced by TPA or Poly(I:C) and belongs to pathological tissue or cells; while here, the human keratinocytes only represent the normal skin cells. Why did not investigate the potential effect of Phen on signaling pathways and inflammatory factors in keratinocytes after treatment with Poly(I:C)?

6. Figure 4, The expression of c-myc needs to be evaluated in the inflammatory models.

7. The roles of c-myc in inflammation should be fully discussed in this manuscript because the authors considered that Phen inhibits the expression of the inflammatory factors through control c-Myc expression, but not classical regulatory approaches. 

Round 2

Reviewer 2 Report

In revised manuscript authors addressed the issues raised by the reviewers and updated the manuscript. In my opinion the the revised manuscript can be accepted in the present form.

Reviewer 3 Report

I appreciate that the authors have addressed the comments. The revised version of the manuscript “Phenformin downregulates c-Myc expression to suppress the expression of pro-inflammatory cytokines in keratinocytes” has improved considerably.